∂ | **Open Peer Review** | Clinical Microbiology | Research Article

# Diagnostic value of metagenomic next-generation sequencing using bronchoalveolar lavage fluid samples for pathogen detection in children with severe or refractory pneumonia

Meixi Zhao,[1] Yanyan Shi,[2] Congcong Zhang,[2] Meiqin Lu,[1] Meili Shen,[3] Lijian Xie,[1] Libo Wang,[1] Aizhen Lu[1]

**ABSTRACT** Pneumonia is the leading cause of morbidity and mortality in children and needs rapid and accurate pathogenic diagnosis. The aim of this study was to evaluate the diagnostic value of bronchoalveolar lavage fluid (BALF) metagenomic next-generation sequencing (mNGS) and conventional microbiological tests (CMTs) for pathogen detection in children with severe or refractory pneumonia. In this retrospective study, the clinical data of 127 children with severe or refractory pneumonia admitted to the respiratory department from June 2021 to March 2022 were analyzed. BALF mNGS and CMTs were utilized for pathogen diagnosis and comparison of their detection performance for different pathogens. The pathogenic diagnosis rate was 95.28% (121/127) by combining mNGS and CMTs. mNGS had significantly higher overall (96.06% vs 72.44%, $P < 0.001$), bacterial (69.29% vs 12.60%, $P < 0.001$), and fungal (11.81% vs 3.15%, $P = 0.009$) detection rates than CMTs. However, there was no significant difference of detection rates between them for respiratory viruses (33.86% vs 33.75%, $P = 0.99$) and *Mycoplasma pneumoniae* (48.03% vs 45.67%, $P = 0.71$). The sensitivities of mNGS for total pathogens, bacteria, and fungi were 99.17%, 100%, and 87.50%, respectively, which were higher than those of CMTs. CMTs for *M. pneumoniae* had the highest sensitivity (91.23%) compared with mNGS (89.47%) and multiplex PCR (88.57%). For respiratory viruses, mNGS and mPCR had similar sensitivities (97.67% vs 96.43%). mNGS was superior to CMTs in bacterial and fungal detection, while it was comparable to multiplex PCR for the detection of *M. pneumoniae* and respiratory viruses. Different detection methods should be rationalized for different pathogens.

**IMPORTANCE** This study on 127 patients with severe and refractory pneumonia showed that mNGS was significantly superior to CMTs in terms of bacterial and fungal detection. We also found that multiplex PCR assay was comparable to mNGS for the detection of *Mycoplasma pneumoniae* and respiratory viruses and may have greater application advantages in combination with CMTs, such as *M. pneumoniae* IgM. For severe and refractory pneumonia, or when empiric treatment is not effective, collecting BALF for mNGS can help to quickly identify the causative organisms at an early stage. It is also important to choose more appropriate methods or combinations for different pathogens.

**KEYWORDS** severe or refractory pneumonia, metagenomic next-generation sequencing, children, *Mycoplasma pneumoniae*, pathogen detection

P neumonia is the leading cause of morbidity and mortality in children, especially those <5 years of age (1, 2). In 2019, pneumonia-related deaths ($n$ = 740,180) accounted for 14% of all deaths in children aged <5 years (3). Refractory pneumonia is a type of pneumonia in which symptoms either do not improve, continue to

**Peer Reviewer** Anne Jamet, Institut Necker Enfants Malades, Paris, France

Address correspondence to Lijian Xie, naijileix@aliyun.com, Libo Wang, wanglbc@163.com, or Aizhen Lu, zal2008@163.com.

The authors declare no conflict of interest.

worsen, or prolong the course of the disease despite active management, including anti-infection treatments (4). Unidentified infections and co-infections are a major cause of poor empirical treatment outcomes (5). Current conventional microbiological tests (CMTs) have limitations in detection range, positive rates, and turnaround time, which hinder their ability to meet the demand for rapid and accurate diagnosis of pneumonia pathogen diagnosis. Metagenomic next-generation sequencing (mNGS) offers advantages in terms of turnaround time and sensitivity and has already demonstrated its value in clinical infection diagnosis (5–9). The use of mNGS has also been reported in several studies in the diagnosis of pediatric pneumonia (10–15), but the number of patients enrolled is still relatively small, and RNA pathogen testing is either not or incompletely covered. For different infected populations and pathogen types, the decision to include mNGS or CMTs alone also requires the accumulation of more real-world data.

In this study, 127 children with refractory and severe pneumonia were retrospectively enrolled, and bronchoalveolar lavage fluid (BALF) samples were used for mNGS testing at both DNA and RNA levels, which was combined with CMTs for comprehensive pathogen diagnosis of pneumonia. We also compared the performance of mNGS and CMTs in detecting different pathogens, providing valuable suggestions for screening various combinations of pathogen detection methods in pneumonia.

## MATERIALS AND METHODS

### Patient enrollment and study design

We screened cases diagnosed with severe or refractory pneumonia from 160 patients who underwent BALF mNGS. A total of 127 pediatric patients with severe or refractory pneumonia admitted to the respiratory department of the Children's Hospital of Fudan University between June 2021 and March 2022 were retrospectively enrolled in this study, and their clinical data including BALF mNGS and CMTs were collected (Fig. 1).

The enrollment criteria were as follows: severe pneumonia was diagnosed by the presence of any of the following conditions along with the pneumonia: (i) poor general condition (ii); refusal of food or signs of dehydration (iii); impaired consciousness (iv); markedly increased respiratory rate (v); cyanosis (vi); respiratory distress (moaning, nasal flaring) (vii); multilobar involvement of the lungs or ≥2/3 pulmonary infiltrates (viii); pleural effusion (ix); pulse oximetry ≤92%; and (x) extrapulmonary complications. Pneumonia was considered refractory if it did not improve after more than 15 days of conventional treatment (4). The exclusion criteria were as follows: (i) clinical data were incomplete, and (ii) BALF-mNGS or CMTs were not performed. The study was performed in accordance with the Declaration of Helsinki and was approved by the ethics committee of the Children's Hospital of Fudan University (approval ID: 2022–239).

### Sample collection

Bronchoalveolar lavage was performed under sedation and local anesthesia after obtaining indication and informed consent. Lavage was preferably performed in the diseased lung segment, or in the case of diffuse infection, in the middle lobe of the right lung or the lingual lobe of the left lung. Briefly, 1–3 mL/kg sterile saline was used for lavage, and BALF was aspirated under negative pressure into a sterile collector. BALF samples were then immediately processed and stored according to the requirements of the laboratory's CMTs and mNGS. Peripheral blood was also collected for CMTs.

### Pathogen detection

BALF mNGS based on DNA and RNA were performed in a certified and accredited laboratory of Nanjing Dinfectome Technology Inc., Nanjing, Jiangsu, China. BALF DNA was extracted using the TIANamp Magnetic DNA Kit (TIANGEN, China), and BALF RNA was extracted using the QIAamp Viral RNA Mini Kit (QIAGEN, Germany) according to the

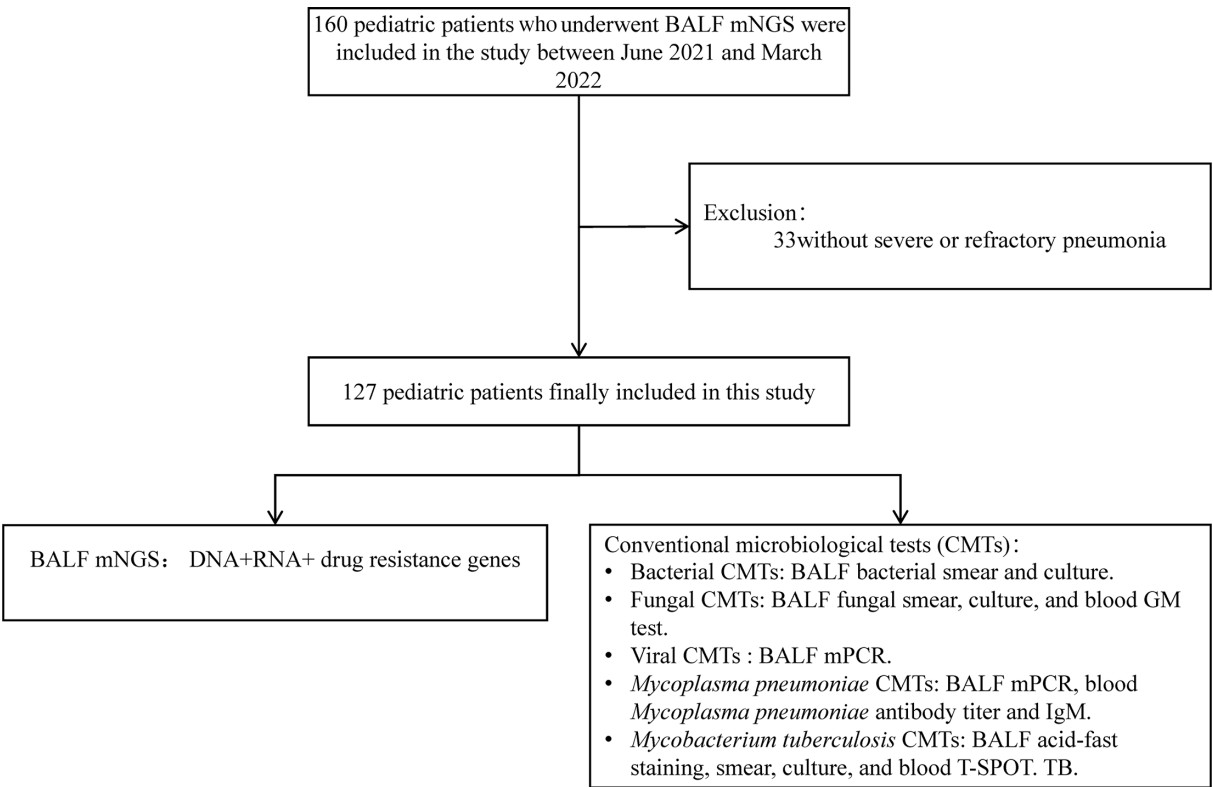

**FIG 1** Workflow of patient inclusion and exclusion.

manufacturer's protocol. After quantitative and qualitative evaluation of DNA and RNA, a DNA library was constructed using Hieff NGS OnePot II DNA Library Prep Kit (Yeasen Biotech, Shanghai, China), and an RNA library was constructed by the VAHTS Universal V8 RNA-seq Library Prep Kit (Vazyme Biotech, China), according to the manufacturers' protocol. Finally, we completed sequencing in the single-end 50 bp sequencing mode using DIFSEQ-200 (Dinfectome, China). No template negative controls (NTCs) were included in the extraction, library preparation, and sequencing process. Raw sequencing data were split by bcl2fastq2, and high-quality sequencing data were generated using Trimmomatic by removing low-quality, adapter contaminated, duplicated, and short (<36 bp) reads. Human host sequences were subtracted by mapping to the human reference genome (hs37d5) using bowtie2. Reads that could not be assigned to the human genome were retained and aligned with the microorganism genome database for microbial identification using Kraken 2, and the species abundance were estimated by Bracken. The microorganism genome database containing genomes or scaffolds of bacteria, fungi, viruses, and parasites were downloaded from GenBank (release 238, https://ftp.ncbi.nlm.nih.gov/genomes/genbank/). The background of microbial communities occurring in the normal population was deducted to identify the potential pathogenic microorganisms.

Conventional microbiological tests included smear, acid-fast staining (AFS) and BALF culture, multiplex PCR (mPCR) for 13 pathogens of BALF, including influenza A virus, adenovirus, bocavirus, rhinovirus, influenza A virus H1N1, parainfluenza virus, *Chlamydia*, metapneumovirus, influenza B virus, *Mycoplasma pneumoniae* (MP), seasonal H3N2 virus, coronavirus, and respiratory syncytial virus, and blood MP IgM (indirect immunofluorescence assay) and MP IgM antibody titer (passive agglutination method), blood tuberculosis infection T-cell spot (T-spot), and blood galactomannan (GM) test.

## Pathogen diagnosis

Owing to the lack of standard methods for interpreting mNGS results and the diversity of reported parameters among different sequencing platforms, the following criteria were used to define clinically significant microorganisms (CSMs) based on the mNGS results (16): (i) For bacteria (excluding *Mycobacterium*), fungi (excluding molds), viruses, and parasites, CSM was classified if the relative abundance of a microorganism at the species level exceeded 30% and if that microorganism was previously found to be pathogenic. A microorganism can be categorized as a CSM if it is detected as a major pathogen at both the DNA and RNA levels, even if its relative abundance is <30% (ii). Owing to the low wall-breaking extractions rate of *Mycobacterium*, if the number of strictly aligned sequences at the species level was ≥1, and contamination was ruled out using NTCs, the microorganism was classified as a CSM. This also applied to *Mucor*, *Cryptococcus*, and *Yersinia pestis* (iii). Microorganisms detected in the NTCs were excluded unless the sequence number was ≥10-fold higher than that in the NTCs. The mNGS results cannot be used to determine whether a microorganism is infecting, colonizing or contaminating. We will also review the original mNGS detection data if necessary. There are no standardized criteria to establish the causative agent of infection. Generally, it needed to be highly considered if a microorganism was detected by both CMTs and mNGS. For patients without CMTs or with negative results, clinical diagnosis can be made when clinical improvement was achieved after adjusting the treatment regimen based on mNGS results. Pathogen diagnosis was determined by two experienced clinicians based on epidemiology, clinical presentation, treatment outcome, laboratory findings, chest radiology findings, and the host's immune status, using a combination of mNGS and CMTs clinical diagnostic criteria. If two clinicians were unable to reach an agreement, an in-depth discussion was carried out with another senior expert to reach a consensus.

## Statistical analysis

Continuous variables were expressed as the mean ± standard deviation (SD), and comparisons between groups were performed using Student's *t*-test. Categorical variables were expressed as percentages (%), and statistical tests were performed using the chi-squared test or Fisher's exact test. All statistical calculations were performed using SPSS 24.0 statistical software (IBM Corporation, Armonk, NY, USA). $P < 0.05$ was considered to indicate statistical significance.

Following the extracted data, 2 × 2 contingency tables were established to calculate sensitivity, specificity, positive predictive value (PPV), negative predictive value (NPV), and accuracy of mNGS and CMTs methods, respectively. Pathogen diagnosis was used as the reference standard.

## RESULTS

### Patient enrollment and clinical characteristics

This study enrolled 127 children (67 girls and 60 boys) with severe and refractory pneumonia (Table 1). The mean age of the patients was 5.14 ± 4.18 years, and the mean hospital stay was 8.72 ± 3.55 days. Nineteen children had one or more underlying diseases, including growth retardation (*n* = 2), chromosomal abnormalities (*n* = 2), tracheomalacia (*n* = 3), cystic fibrosis (*n* = 4), primary ciliary dyskinesia syndrome (*n* = 3), pulmonary hemosiderosis (*n* = 3), bronchopulmonary dysplasia (*n* = 1), and aspiration pneumonia (*n* = 4). One patient was referred to a sentinel hospital for further tuberculosis treatment; the remaining 126 showed good clinical regression. The data of 127 pediatric patients are presented in Table S1.

### Pathogen diagnosis and distribution

The pathogen diagnosis rate was 95.28% (121/127) in 127 children with severe and refractory pneumonia (Fig. 2). A total of 185 strains of 27 pathogens were detected in

**TABLE 1** Baseline characteristics of 127 pediatric patients with severe or refractory pneumonia[a]

| Characteristics | Count (%)/mean ± SD |
| --- | --- |
| Sex (male) | 60 (47.24) |
| Age (years) | 5.14 ± 4.18 |
| Underlying disease | 19 (14.96) |
| Growth retardation | 2 (1.57) |
| Chromosomal abnormalities | 2 (1.57) |
| Tracheomalacia | 3 (2.36) |
| Cystic fibrosis | 4 (3.15) |
| Primary ciliary dyskinesia syndrome | 3 (2.36) |
| Pulmonary hemosiderosis | 3 (2.36) |
| Bronchopulmonary dysplasia | 1 (0.79) |
| Aspiration pneumonia | 4 (3.15) |
| Hospital stays (day) | 8.72 ± 3.55 |
| Outcome (cured) | 126 (99.21) |
| Antibiotic adjustments | 30 (23.62) |
| Refer to mNGS results | 26 (20.47) |
| Refer to culture and drug sensitivity test | 3 (1.00) |
| Refer to MP antibody titer | 2 (1.57) |

[a]mNGS: metagenomic next-generation sequencing; MP: *Mycoplasma pneumoniae*.

121 children (Fig. 3), including rare cases of *Pneumocystis jirovecii* (1), *Rhizopus microsporus* (1), *Bordetella parapertussis* (1) and *M. tuberculosis* (1). Among them, *Haemophilus parainfluenzae*, *R. microspores*, *P. jirovecii*, *Tropheryma whipplei*, *Streptococcus pseudopneumoniae*, *B. parapertussis*, *Penicillium digitatum*, *Candida albicans*, and human betaherpesvirus 5 (Cytomegalovirus, CMV) were only detected by mNGS. *Enterobacter cloacae* was only detected by CMTs. The pathogen detection rates were 47.24% (60/127), 44.88%

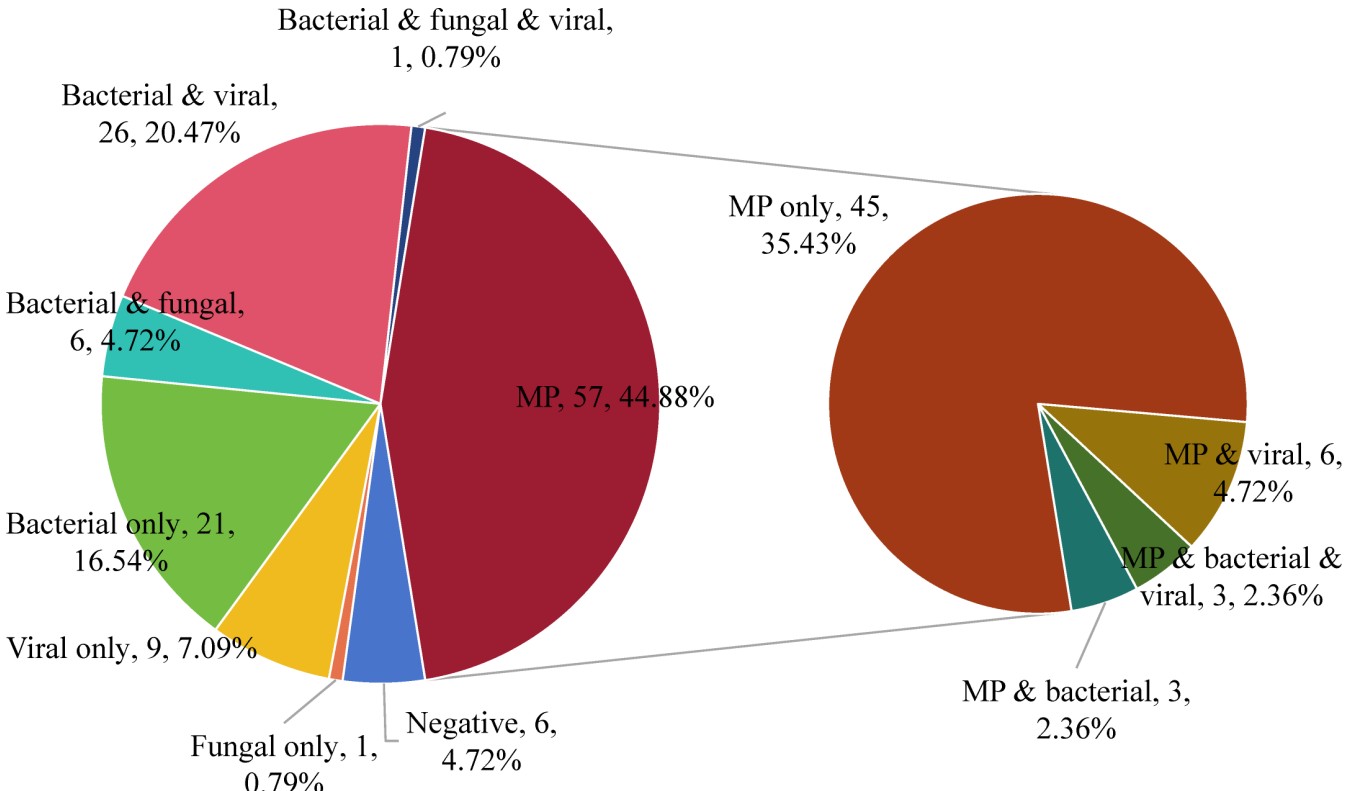

**FIG 2** Distribution of pathogen diagnosis in 127 pediatric patients with severe or refractory pneumonia. MP: *Mycoplasma pneumoniae*.

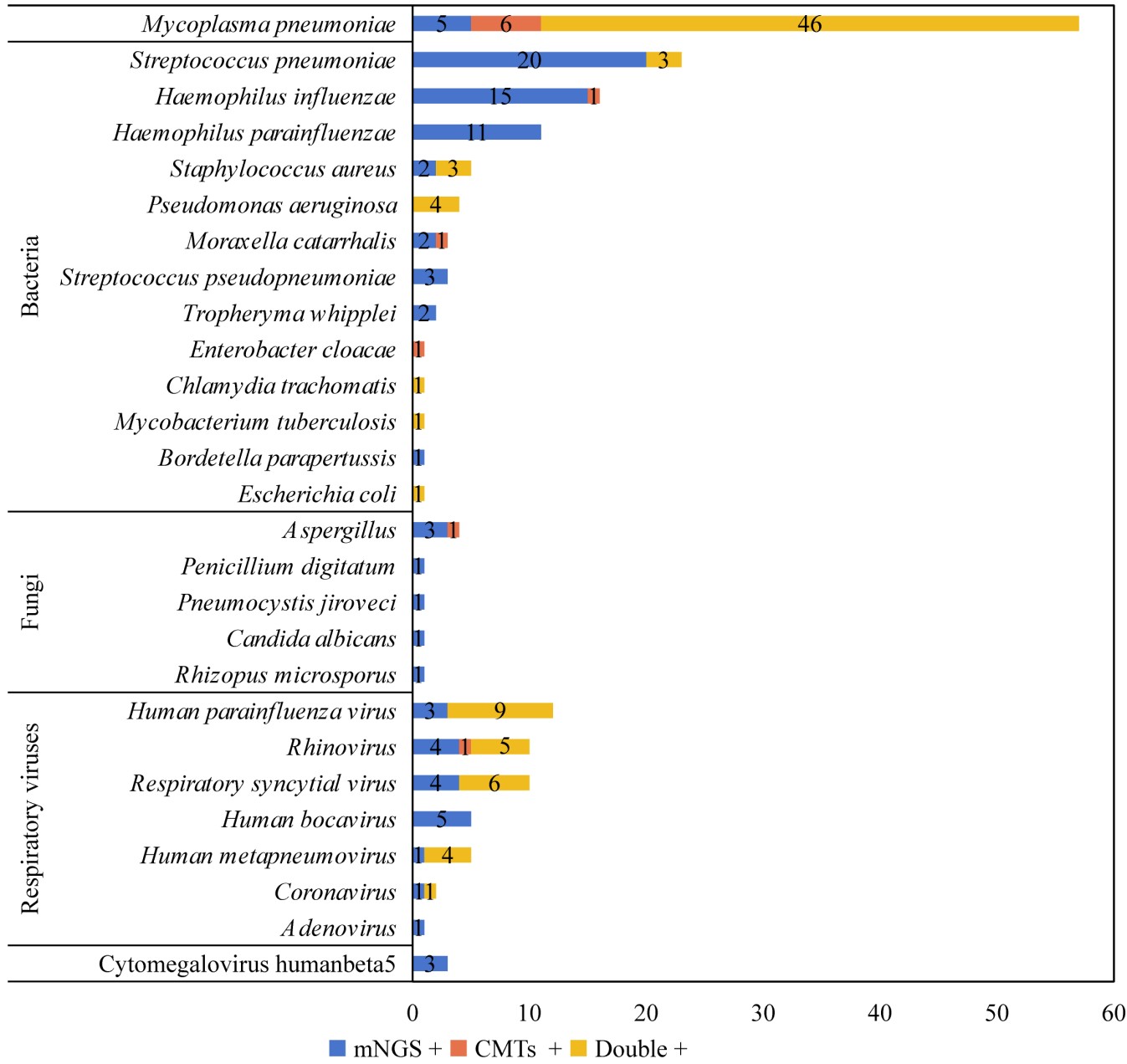

**FIG 3** Comparison of pathogen detection between mNGS and conventional microbiological tests. mNGS: metagenomic next-generation sequencing; CMTs: conventional microbiological tests.

(57/127), 6.30% (8/127), 33.86% (43/127), and 2.36% (3/127) for bacteria (excluding *M. pneumoniae*), *M. pneumoniae*, fungi, respiratory viruses, and CMV, respectively.

The type of infection was categorized into simple and mixed infections according to the combination of pathogens (Fig. 2). The most common simple infections were with *M. pneumoniae* (45/127), followed by bacterial infections (21/127). Most forms of mixed infections were bacterial or *M. pneumoniae* combined with viral and/or fungal infections, with bacterial combined with viral being the most common type (26/127). Of the 12 mixed infections with *M. pneumoniae*, patients infected with *M. pneumoniae* in combination with viral, bacterial, and bacterium–viral infections were 6, 3, and 3, respectively.

## Diagnostic performance of mNGS and CMTs

The overall pathogen detection rate of mNGS was higher than that of CMTs (96.06% vs 72.44%, $P < 0.001$; Fig. 4A). The detection rates of mNGS for both bacteria (69.29% vs 12.60%, $P < 0.001$) and fungi (11.81% vs 3.15%, $P = 0.009$) were significantly higher than those of CMTs (Fig. 4A). However, there was no statistically significant difference in the comparison of detection rates between mNGS and CMTs for respiratory viruses (33.86% vs 33.75%, $P = 0.99$) and MP (48.03% vs 45.67%, $P = 0.71$). CMTs for MP include IgM,

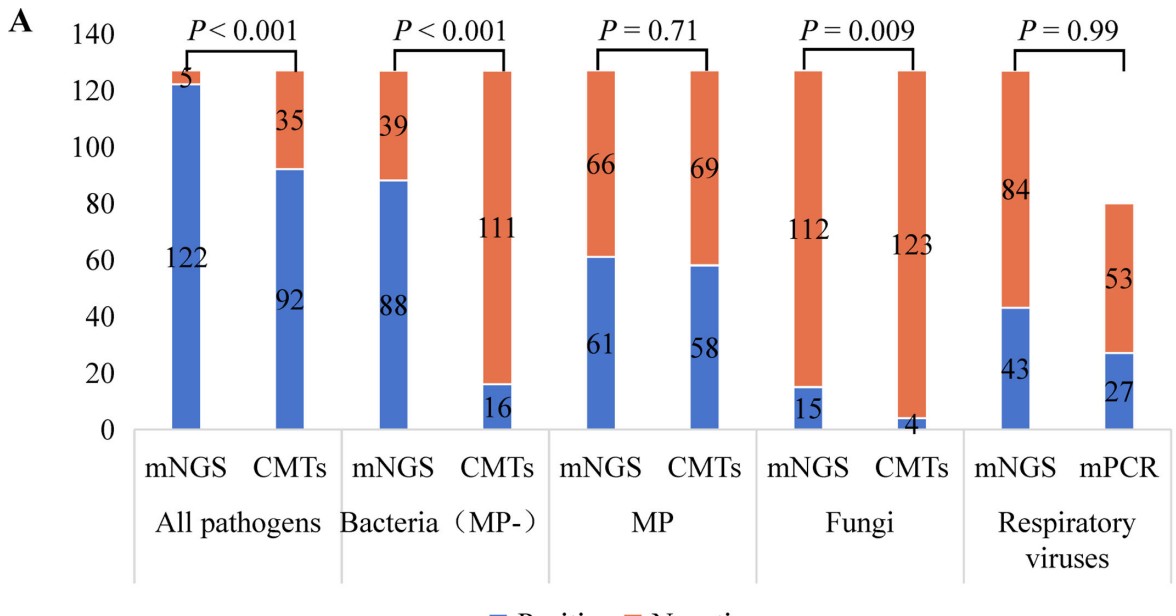

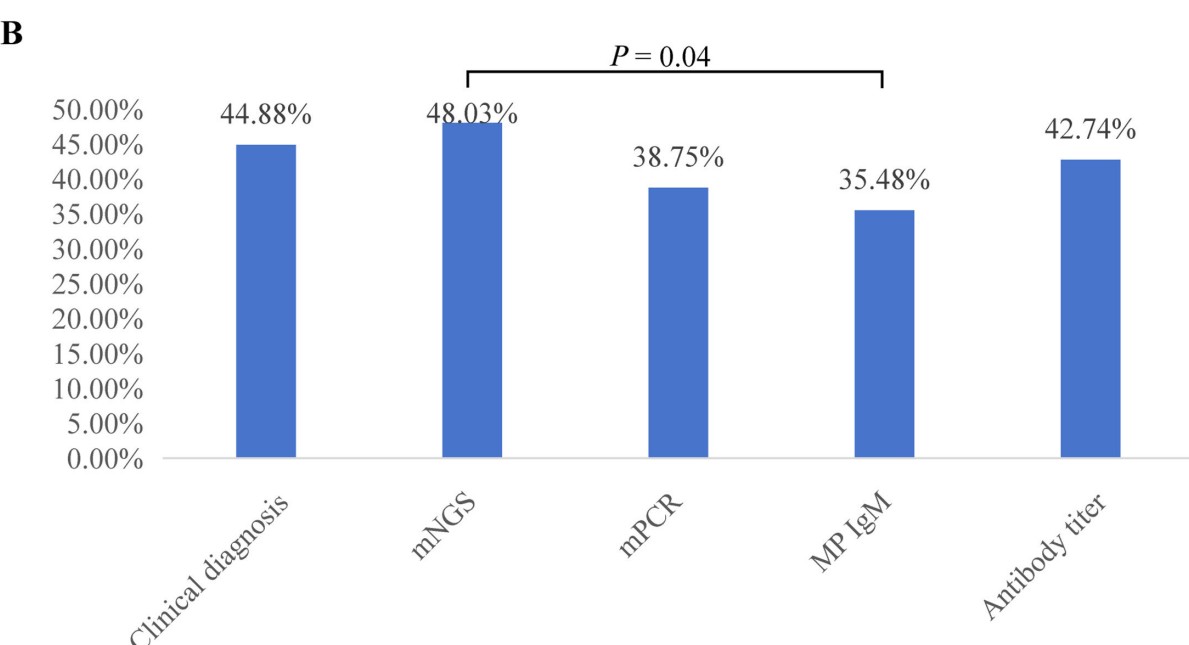

**FIG 4** Comparison of positive detection rates for various pathogens and detection methods. (A) The positive rates of all pathogens, bacteria (excluding *M. pneumoniae*), *M. pneumoniae*, fungi, and respiratory viruses detected by mNGS and CMTs. (B) The positive rates of *M. pneumoniae* diagnosed and detected by mNGS, mPCR, IgM, and IgM antibody titer. The χ test was used to test for differences in positivity rates among the four methods, specifically in a two-by-two comparison. The results showed significant differences only between the mNGS and IgM methods ($P = 0.04$). mNGS: metagenomic next-generation sequencing; CMTs: conventional microbiological tests; mPCR: multiplex PCR.

IgM antibody titer, and mPCR. mNGS had a significantly higher positive detection rate than the IgM method (48.03% vs 35.48%, *P* = 0.04), but not higher than other methods (Fig. 4B). Out of 57 patients with *M. pneumoniae* pneumonia (MPP) in this study, 11 were diagnosed with refractory *M. pneumoniae* pneumonia (RMPP). Resistance gene mutation detection was also performed using mNGS. Among these, four were found to have resistance gene mutation, 23s rRNA gene A2063G. The same resistance gene mutation was detected in 12 out of 46 patients with common MPP. The detection rate between the two groups did not show a significant difference (36.4% vs 25%, *P* = 0.697; Table S1).

We also calculated the sensitivity, specificity, positive predictive value (PPV), negative predictive value (NPV), and accuracy of mNGS and CMTs, respectively (Table 2). For bacteria except for *MP*, the sensitivity, specificity and accuracy of mNGS were 100.00%, 58.21%, and 77.95%, respectively, while the sensitivity, specificity and accuracy of CMTs were 26.67% (*P* < 0.001), 100.00% (*P* < 0.001), and 65.35% (*P* = 0.026), which was significantly different from mNGS. The PPV and NPV of mNGS for bacteria were 68.18% and 100.00%, respectively. For fungi, mNGS showed a higher sensitivity of 87.50% compared with CMTs of 12.50% (*P* = 0.01), while the specificity of mNGS was close to that of CMTs (95.28% vs 97.48%, *P* = 0.12). PPV, NPV, and accuracy were not significantly different between mNGS and CMTs. For the detection of MP, mPCR exhibited higher specificity (100.00% vs 85.71%) and accuracy (95.00% vs 87.40%) than mNGS, while the sensitivity decreased (88.57% vs 89.47%). MP CMTs had the highest sensitivity (91.23%), followed by mNGS (89.47%) and mPCR (88.57%). But these parameters were not significantly different between mNGS and CMTs. For respiratory viruses, mNGS and mPCR showed similar sensitivity (97.67% vs 96.43%), while mPCR demonstrated slightly higher specificity (100.00% vs 98.81%) and accuracy (98.75% vs 98,43%). The PPV and NPV of mPCR for respiratory viruses were 100.00% and 98.11%, respectively. All these parameters were not significantly different between mNGS and CMTs.

## Antibiotic adjustment referred to pathogen diagnosis

Antibiotics were adjusted based on pathogen detection results in 30 cases (Table S1). Of these, 26 cases changed antibiotics according to mNGS, resulting in an improvement for 24 patients. Pathogen detected by mNGS included *M. pneumoniae*, *H. parainfluenzae*, *Streptococcus pneumoniae*, *Staphylococcus aureus*, *H. influenzae*, *T. whipplei*, *E. coli*, *P. aeruginosa*, Cytomegalovirus humanbeta5, rhinovirus, *Aspergillus*, *R. microsporus*, *P. jirovecii*, and *C. albicans*. The remaining two cases were treated with linezolid to

**TABLE 2** Diagnostic performance of mNGS and CMTs in pediatric patients with severe or refractory pneumonia[d]

| Pathogens | Methods | Sensitivity, % (95% CI) | Specificity, % (95% CI) | PPV, % (95% CI) | NPV, % (95% CI) | Accuracy, %(95% CI) |
|---|---|---|---|---|---|---|
| All pathogens | mNGS | 99.17 (95.48, 99.98) | 66.67 (22.28, 95.67) | 98.36 (94.20, 99.80) | 80.00 (28.36, 99.49) | 97.64 (93.25, 99.51) |
| | CMTs | 76.03 (67.43, 83.32) | 100.00 (54.07, 100.00) | 100.00 (96.07, 100.00) | 17.14 (6.56, 33.65) | 77.17 (68.88, 84.14) |
| Bacteria (MP-) | mNGS | 100.00 (94.04, 100.00) | 58.21 (45.52, 70.15) | 68.18 (57.39, 77.71) | 100.00 (90.97, 100.00) | 77.95 (69.74, 84.82) |
| | CMTs[a] | 26.67 (16.07, 39.66) | 100.00 (94.64, 100.00) | 100.00 (79.41, 100.00) | 60.36(50.63, 69.52) | 65.35 (56.40, 73.57) |
| MP | mNGS | 89.47 (78.48, 96.04) | 85.71 (75.29, 92.93) | 83.61 (71.91,91.85) | 90.91 (81.26, 96.59) | 87.40 (80.35, 92.62) |
| | CMTs[b] | 91.23 (80.70, 97.09) | 91.43 (82.27, 96.79) | 89.66 (78.83, 96.11) | 92.75 (83.89, 97.60) | 91.34 (85.03, 95.60) |
| | mPCR | 88.57 (73.26, 96.80) | 100.00 (92.13, 100.00) | 100.00 (88.78, 100.00) | 91.84 (80.40, 97.73) | 95.00 (87.69, 98.62) |
| | IgM | 66.07 (52.19, 78.19) | 89.70 (79.93, 95.76) | 84.09 (69.93, 93.36) | 76.25 (65.42, 85.05) | 79.03 (70.81, 85.82) |
| | Antibody titer | 78.57 (65.56, 88.41) | 90.16 (79.81, 96.30) | 88.00 (75.69, 95.47) | 82.09 (70.80, 90.39) | 84.62 (76.78, 90.62) |
| Fungi | mNGS | 87.50 (47.35, 99.68) | 93.28 (87.18, 97.05) | 46.67 (21.27, 73.41) | 99.11 (95.13, 99.98) | 92.91 (86.97, 96.71) |
| | CMTs[c] | 12.50 (0.32, 52.65) | 97.48 (92.81, 99.48) | 25.00 (0.63, 80.59) | 94.31 (88.63, 97.68) | 92.13 (86.00, 96.16) |
| Respiratory viruses | mNGS | 97.67 (87.71, 99.94) | 98.81 (93.54, 99.97) | 97.67 (87.71, 99.94) | 98.81 (93.54, 99.97) | 98.43 (94.43, 99.81) |
| | mPCR | 96.43 (81.65, 99.91) | 100.00 (93.15, 100.00) | 100.00 (87.23, 100.00) | 98.11 (89.93, 99.95) | 98.75 (93.23, 99.97) |

[a]CMTs for bacteria included culture, smear, and acid-fast staining.
[b]CMTs for MP included mPCR, IgM, and antibody titer.
[c]CMTs for fungi included culture, smear, and GM test.
[d]MP: *Mycoplasma pneumoniae*; mNGS: metagenomic next-generation sequencing; CMTs: conventional microbiological tests; mPCR: multiplex PCR; GM test: galactomannan test; PPV: positive predictive value; NPV: negative predictive value.

target *Enterococcus* infections, guided by the results of mNGS. However, there was no notable improvement, and no enterococcal infection was confirmed. In three cases, the antibiotics were changed based on culture and drug sensitivity test, resulting in an improvement in all. In two cases, azithromycin was added based on MP IgM antibody titer, resulting in an improvement for one patient.

## DISCUSSION

By combining clinical features, mNGS, and CMTs, an etiologic diagnosis was achieved in 95.28% (121/127) cases in this study, which is similar to the previously reported rate of >90% (11, 15). Bacteria were found to be the most common cause of pneumonia in children, with a prevalence rate of 47.24% (60/127). The most common bacterial infections were *S. pneumoniae* (23/60), *H. influenzae* (16/60), and *H. parainfluenzae* (11/60), which is consistent with previous findings (13). The study identified parainfluenza virus (12/45), respiratory syncytial virus (10/45), and rhinovirus (10/45) as the top three respiratory viruses. These viruses were also reported to be the most prevalent in mainland China (17). Furthermore, respiratory viruses are most frequently found in mixed infections, especially viral and bacterial infections. There is a correlation between mixed viral–bacterial infection patterns and severe pneumonia, for the interaction between viruses and bacteria can promote each other's development, leading to recurrent or prolonged illness (18). Therefore, in addition to anti-infective treatment, prevention of cross-infection and enhancement of the body's immunity are essential in clinical practice. The incidence of *M. pneumoniae* pneumonia (MPP) is increasing annually, with epidemics occurring every 3–5 years, and the age of onset gradually decreasing. Refractory *M. pneumoniae* pneumonia (RMPP), in particular, can elicit a robust immune response and cause severe lung lesions, leading to severe pneumonia. The identification of RMPPs may benefit from detecting drug-resistant mutations in *M. pneumoniae* (11). Out of 57 patients with MPP in this study, the drug-resistant mutation rate of *M. pneumoniae* between the RMPP and common MPP groups did not show a significant difference (36.4% vs 25%, $P = 0.697$). However, it has been suggested that the occurrence of RMPP is not significantly correlated with mutations in *Mycoplasma* resistance genes, rather with a highly homogeneous microbial community dominated by *M. pneumoniae* in BALF (19). Thus, it is essential to determine RMPP in conjunction with a thorough clinical evaluation.

For bacteria except *MP*, mNGS showed higher sensitivity (100.00%) and accuracy (77.95%) than CMTs, with a PPV and NPV of 68.18% and 100.00%, respectively. A negative mNGS result can generally be considered reliable for excluding infections from most bacteria. However, for *Mycobacterium tuberculosis*, the challenge of breaking the cell wall can result in false-negative mNGS test outcomes. Therefore, these results should be interpreted alongside other factors, such as clinical manifestations and laboratory tests. It is also crucial that the test specimen is representative of the site of infection. These findings are consistent with those of other studies (12). However, it was also shown that the bacterial detection rate and sensitivity of mNGS is significantly lower than that of CMTs (15), which may because of the diversity of specimen types used for culture and CMTs and the higher sampling frequency in the PICU. Traditional bacterial cultures often yield low positive rates. Some studies have demonstrated that mNGS can enhance the detection rate of a wide range of bacteria, including *S. pneumoniae*, *H. influenzae*, and *E. coli* (13, 20). This is beneficial for early diagnosis and treatment. Several studies have shown the value of mNGS in fungal detection (12, 15, 21). In this study, mNGS demonstrated a substantially higher detection rate for fungi than CMTs. However, the impact of mNGS only was also found to be limited. In our study, a patient (case 104 in Table S1) who tested negative for mNGS and culture was diagnosed with *Aspergillus* infection by the GM test and 1,3-β-D glucan (G) test (outer hospital testing) and showed a notable improvement in response to voriconazole antifungal therapy. In another study, a patient who initially tested negative for fungus on both mNGS and culture was later diagnosed with a fungal infection by the G test (13). These findings suggest that a combination of culture, mNGS, G test, and GM test is preferred for the diagnosis of fungal

infections despite the large advantage of mNGS. While mNGS has obvious advantages in detecting bacteria and fungi, it is better suited for screening uncommon and rare pathogens for which there are limited conventional assays. The uncommon pathogens require additional attention for screening in at-risk children like immunocompromised or with severe pneumonia (22–24). Children with CMV pneumonia in this study were aged 2, 4, and 5 months and had clinical manifestations of severe pneumonia. *Pseudomonas aeruginosa* and fungal infections were diagnosed in 3 out of 4 children with cystic fibrosis. Children with aspiration pneumonia may have underlying conditions, such as neurological or respiratory abnormalities (25) and are susceptible to infection with bacteria, such as *E. coli* and *P. aeruginosa*, as demonstrated in this study. Among all diagnosed pathogens in this study, *H. parainfluenzae*, *R. microsporus*, *P. jirovecii*, *T. whipplei*, *S. pseudopneumoniae*, *Bordetella parapertussis*, *Penicillium digitatum*, *C. albicans*, and Cytomegalovirus *humanbeta5* were only detected by mNGS. In another study (26), *P. jirovecii* was detected in nearly 15% children aged <6 months, which can only be detected through mNGS. Therefore, for severe pneumonia, early application of mNGS screening for bacterial infection can avoid duplication and overdetection and provide recommendations for treatment improvement. In this study, 24 children experienced significant therapeutic efficacy and clinical improvement after their antibiotics changed based on the mNGS results. The results of mNGS have not been able to provide an accurate explanation of colonization and infection. *H. parainfluenzae*, previously thought to be a respiratory commensal (27), was also found to be pathogenic in this study according the improvement after antibiotic adjustment and in another study (11). *H. parainfluenzae* may be the pathogen responsible for the disease in immunocompromised patients without other evidence of etiology, especially when antibiotics are not effective (28). In some cases, the pathogenic presumption was based on the detection of active expression in both DNA and RNA. This suggests that we also need to analyze mNGS results for rare pathogens in conjunction with CMTs and the patient's immune status to avoid over-interpretation.

We noted no significant difference in detection rate of mNGS compared to CMTs for *MP* and respiratory viruses. For the detection of MP, mPCR exhibited higher specificity (100.00% vs 85.71%) and accuracy (95.00% vs 87.40%) than mNGS, while the sensitivity decreased (88.57% vs 89.47%). CMTs had the highest sensitivity (91.23%), followed by mNGS and mPCR. This finding is consistent with previous studies (13); however, another study showed that mNGS was superior to conventional methods in detecting *M. pneumoniae* (14). This may be because of factors, such as the patient population and the combination of conventional methods used for *MP* detections. In this study, CMTs for *MP* included IgM and IgM antibody titer and mPCR. mNGS had a significantly higher positive detection rate than the IgM method (48.03% vs 35.48%, $P = 0.04$) but was not higher than other methods. Considering the biological characteristics, we need to consider the time window of antibody production when interpreting IgM results. Other methods, such as nucleic acid tests, are needed to provide evidence as well. In this study, mNGS, IgM, and antibody titer were all tested in the acute phase of the infection. This may be why mNGS had a higher positivity rate than IgM. For respiratory viruses, mNGS and mPCR showed similar sensitivity (97.67% vs 96.43%), while mPCR demonstrated slightly higher specificity (100.00% vs 98.81%) and accuracy (98.75% vs 98,43%). The PPV and NPV of mPCR for respiratory viruses were 100.00% and 98.11%, respectively. The mPCR used in this study could detect 13 common respiratory pathogens at one-third the cost of mNGS, making it more suitable for screening of common respiratory viruses and MP.

This study has some limitations. The retrospective analysis was limited to the detection and diagnostic techniques available at the time and may have been biased. We have tried to be as impartial and standardized as possible in determining the final pathogen, but it may still be impossible to avoid some degree of bias. Some of the conclusions drawn from this study need to be validated by further increasing the sample size.

In conclusion, this study on 127 patients with severe and refractory pneumonia showed that mNGS was significantly superior to CMTs in terms of bacterial and fungal detection. For severe and refractory pneumonia, or when empiric treatment is not effective, collecting BALF for mNGS can help to quickly identify the causative organisms at an early stage. Although there are increasing reports of the successful application of mNGS, several limitations need to be addressed, such as differentiation between colonization and infection, extraneous sources of nucleic acid, and standardization of methods and quality controls (29). The interpretation of mNGS results needs to take into account the age of the host, presence or absence of underlying diseases, and immune status, and be confirmed by routine clinical methods to improve the accuracy of interpretation, while ensuring appropriate sample selection and controlled process quality. We also found that multiplex PCR assay was comparable to mNGS for the detection of *M. pneumoniae*, and respiratory viruses and may have greater application advantages in combination with *Mycoplasma* antibody and viral antigen detection in clinical practice. Various combinations of mPCR assays for respiratory viruses and common bacteria have appeared on the market. Recently, targeted NGS (tNGS) technology is also being used (30), and we believe that in the future, an increasing number of excellent detection methods can be applied clinically to ultimately realize the accurate and rapid diagnosis of infectious etiology.

## ACKNOWLEDGMENTS

Thanks to the Respiratory Department of Fudan Affiliated Pediatric Hospital for providing information and assistance as an assisting department.

This work was supported by 2019 Shanghai Medical Key Specialty Pediatric Construction Plan (ZK2019A010), Seventh cycle Jinshan District Medical Key Specialty Class A – Pediatrics (JSZK2023A04), and Shanghai Medical and Health Technology Innovation Fund Project (2023-WS-14).

M.Z.: Conceptualization, Methodology, Funding acquisition, Investigation, Data curation, Writing-original draft, Visualization, Writing-review & editing; Y.S., C.Z.: Data curation; M.L.: Methodology; M.S.: Writing-original draft, Visualization, Writing-review & editing; L.X.: Supervision; L.W.: Funding acquisition; A.L.: Conceptualization, Supervision, Validation.

Meili Shen is an employee of Nanjing Dinfectome Technology Inc, China.

## AUTHOR AFFILIATIONS

[1]Department of Pediatrics, Jinshan Hospital, Fudan University, Shanghai, China
[2]Department of Respiratory, Children's Hospital of Fudan University, Shanghai, China
[3]Medical Department, Nanjing Dinfectome Technology Inc, Nanjing, China

## AUTHOR ORCIDs

Meixi Zhao http://orcid.org/0009-0004-4414-6172
Lijian Xie http://orcid.org/0000-0002-0328-6820
Aizhen Lu http://orcid.org/0000-0002-3057-1061

## AUTHOR CONTRIBUTIONS

Meixi Zhao, Conceptualization, Data curation, Funding acquisition, Investigation, Methodology, Visualization, Writing – original draft, Writing – review and editing | Yanyan Shi, Data curation | Congcong Zhang, Data curation | Meiqin Lu, Methodology | Meili Shen, Visualization, Writing – original draft, Writing – review and editing | Lijian Xie, Supervision | Libo Wang, Funding acquisition | Aizhen Lu, Conceptualization, Supervision, Validation

## DATA AVAILABILITY

Raw data of 127 cases were deposited in the National Genomics Data Center under project PRJCA028177 (GSA: CRA017789) and are publicly accessible at https://ngdc.cncb.ac.cn/bioproject/browse/PRJCA028177.

## ETHICS APPROVAL

This work was approved by the ethics committee of the Children's Hospital of Fudan University (Approval ID: 2022–239).

## ADDITIONAL FILES

The following material is available online.

### Supplemental Material

**Table S1 : The data of 127 pediatric patients. (Spectrum01087-24-s0001.pdf).**

### Open Peer Review

**PEER REVIEW HISTORY (review-history.pdf).** An accounting of the reviewer comments and feedback.

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
