## [Reviewer comments · Microbiology Spectrum]

Microbiology Spectrum

Diagnostic value of metagenomic next-generation sequencing using bronchoalveolar lavage fluid samples for pathogen detection in children with severe or refractory pneumonia

Meixi Zhao, Yanyan Shi, Congcong Zhang, Meiqin Lu, Meili Shen, Lijian Xie, Libo Wang, and Aizhen Lu

Corresponding Author(s): Meixi Zhao, Jinshan Hospital of Fudan University

Review Timeline:

Submission Date:	April 30, 2024
Editorial Decision:	July 2, 2024
Revision Received:	August 27, 2024
Editorial Decision:	October 26, 2024
Revision Received:	November 11, 2024
Accepted:	December 6, 2024

Editor: Jérôme Le Goff

Reviewer(s): Disclosure of reviewer identity is with reference to reviewer comments included in decision letter(s). The following individuals involved in review of your submission have agreed to reveal their identity: Anne Jamet (Reviewer #2)

Transaction Report:

DOI: <https://doi.org/10.1128/spectrum.01087-24>

Re: Spectrum01087-24 (Diagnostic value of metagenomic next-generation sequencing using bronchoalveolar lavage fluid samples for pathogen detection in children with severe or refractory pneumonia)

Dear Dr. Meixi Zhao:

Thank you for the privilege of reviewing your work. Below you will find my comments, instructions from the Spectrum editorial office, and the reviewer comments.

Revision Guidelines

Sincerely,
Jérôme Le Goff
Editor
Microbiology Spectrum

Reviewer #1 (Public repository details (Required)):

Next gen sequencing information could have been deposited publicly.

Reviewer #1 (Comments for the Author):

Major issues:

- How is the threshold of 30% used validated? It would be good to include a table detailing those findings per case - can be in the supplementary section.
- Line 147 - underlying diseases in 19 patients is only mentioned in the results, the need or correlation of pathogens with comorbidities is unclear. I suggest to remove that or move the table to supplementary data.
- Antibiotic adjustment is quite striking and would merit further discussion and highlight with a table to make it easier to follow.
- Line 185 - I find confusing the statement that "specificity decreased by" for mNGS. Please clarify - what was the specificity calculated from? Clinical diagnosis? from the originally defined 30% threshold? Conventional methods?
- The findings of resistance genes thanks to mNGS should be also highlighted, as it was for MP on line 217.
- Material and methods should include specifically what is included in each CMT described here. Line 245 states that one case failed to detect *Aspergillus* while galactomannan and "Aspergillus antigen" were detected. What antigen test is that? Was that case confirmed to be Aspergillosis and treated like for it?
- Line 248 - what is "G test"
- Line 265 - mNGS shouldn't be compared to detecting IgM titers without adding the context of the biology of IgM, IgM should be discussed together with IgG as IgM is short lived and/or take immunity take longer to mount in an acute infection. Was that MP mNGS positive case, IgM negative also PCR negative?
- Data availability statement states "Data not publicly available", why is it not? NGS sequences could have been made available.

Minor issues:

- Line 62 - correct "form" to "from"
- Line 161 - remove cursive on Cytomegalovirus humanbeta 5
- Consider moving Table 2 to supplementary data

Reviewer #2 (Comments for the Author):

The manuscript by Zhao et al provides interesting information on the diagnostic value of mNGS in the context of children with severe pneumonia by reporting the results of mNGS and CMT in BALF for infection diagnosis in a retrospective cohort of 127 children. The manuscript is clearly written.

I have a few suggestions to improve the manuscript:

Main:

- L100: please specify the databases that have been used since kraken2 and bracken can be used with different databases
- L117: can you provide examples when a 30% cut off is not required beside *Mycobacterium*
- L117: In fact, *Mycobacterium* spp (non-tuberculous species) are possible contaminants in the environment
- L201: the pathogenicity of *H. parainfluenzae* is very low while it is a very common colonizer belonging to the respiratory microbiota DOI: 10.1128/msystems.00178-21. Can you discuss the likelihood of infection with *H. parainfluenzae* in your patients?
- L210-211: can you provide a timeline (or equivalent) of the detected MP to see if there is a link with the season which is an interesting information to guide the use of diagnosis tools
- L231: please be more cautious (at least except *Mycobacteria*)
- L238: "tricky" organisms: cultivating *S. pneumoniae* and *H. influenzae* is definitively less tricky than performing mNGS and allow antibiotic susceptibility testing. I think this really cannot be a justification for the mNGS use...
- L290: tNGS: if possible add a reference

Minor:

- L21: "retrospective" instead of "retrospectively"
- L25: add "by combining" mNGS and CMTs
- L31: add "compared to"
- L41: change by despite active management including anti-infection treatments
- L47: you could add the following reference on mNGS use 10.1016/S2666-5247(23)00244-6
- L62: "from" instead of "form"
- L121: should be "infecting, colonizing or contaminating"
- L151: delete "rate of 100%"
- L151: "for 1 patient" instead of "rate of 50%"
- Fig 3: please use italic when appropriate

Review manuscript: Diagnostic value of metagenomic next-generation sequencing using bronchoalveolar lavage fluid samples for pathogen detection in children with severe or refractory pneumonia

Authors: Zhao M, et al.

Summary: this manuscript evaluates the use of metagenomic next-generation sequencing for pathogen detection in bronchoalveolar lavage and compares it with conventional microbiological tests. The study focuses in children with severe or refractory pneumonia. Due to the general interest in adopting this kind of technologies for the diagnostics of infectious diseases, this study is quite timely.

Major issues:

- How is the threshold of 30% used validated? It would be good to include a table detailing those findings per case – can be in the supplementary section.
- Line 147 – underlying diseases in 19 patients is only mentioned in the results, the need or correlation of pathogens with comorbidities is unclear. I suggest to remove that or move the table to supplementary data.
- Antibiotic adjustment is quite striking and would merit further discussion and highlight with a table to make it easier to follow.
- Line 185 – I find confusing the statement that “specificity decreased by” for mNGS. Please clarify – what was the specificity calculated from? Clinical diagnosis? from the originally defined 30% threshold? Conventional methods?
- The findings of resistance genes thanks to mNGS should be also highlighted, as it was for MP on line 217.
- Material and methods should include specifically what is included in each CMT described here. Line 245 states that one case failed to detect Aspergillus while galactomannan and “Aspergillus antigen” were detected. What antigen test is that? Was that case confirmed to be Aspergillosis and treated like for it?
- Line 248 – what is “G test”
- Line 265 – mNGS shouldn’t be compared to detecting IgM titers without adding the context of the biology of IgM, IgM should be discussed together with IgG as IgM is short lived and/or take immunity take longer to mount in an acute infection. Was that MP mNGS positive case, IgM negative also PCR negative?
- Data availability statement states “Data not publicly available”, why is it not? NGS sequences could have been made available.

Minor issues:

- Line 62 – correct “form” to “from”
- Line 161 – remove cursive on Cytomegalovirus humanbeta 5
- Consider moving Table 2 to supplementary data

Dear Editor Jérôme Le Goff,

Thank you for reviewing our manuscript and allowing us to submit a revised manuscript entitled “Diagnostic value of metagenomic next-generation sequencing using bronchoalveolar lavage fluid samples for pathogen detection in children with severe or refractory pneumonia” (Spectrum01087-24) for further consideration by Microbiology Spectrum.

The authors sincerely appreciate the dedicated time and effort from the reviewers and editors on our manuscript. We are grateful for the insightful comments and have incorporated the reviewers’ suggestions into the revised manuscript. Please refer to the “Response to Reviewers”, in blue, to comments and concerns. We have uploaded the revised main manuscript - “Marked-Up Manuscript” file, where the changes are marked in red and as tracked changes. We also uploaded a clean version of the revised manuscript which convenient for you to browse and proofread the line numbers in this reply.

Thank you for your time and consideration! Please feel free to let us know if you need any further information.

Sincerely,

Meixi Zhao

Jinshan Hospital, Fudan University, Shanghai, China.

Email: 1342940100@qq.com

Point-by-Point Responses

Response to Reviewer 1

Reviewer #1 (Public repository details (Required)):

Next gen sequencing information could have been deposited publicly.

Authors' response:

Thank you for the comment. We have added the NGS data information in "Data availability statement" section (Page 9 Lines 347-350) and in a "Data availability" paragraph placed at the end of the Materials and Methods section (Page 4, Lines 146-149). Please see below and the revised manuscript for details.

"Raw data of 127 cases were deposited in the National Genomics Data Center under project PRJCA028177 (GSA: CRA017789) and are publicly accessible at <https://ngdc.cncb.ac.cn/search/all?&q=PRJCA028177>."

Reviewer #1 (Comments for the Author):

Major issues:

- How is the threshold of 30% used validated? It would be good to include a table detailing those findings per case - can be in the supplementary section.

Authors' response:

Thank you for your suggestion. The threshold of 30% used here was referred to previous study (PMID: 35431561), that was already cited in the text. Interpreting mNGS reports has always been a challenge for clinicians. Here the criteria of clinically significant microorganisms (CSM) were employed to assist clinicians in the

initial screening of mNGS-reporting pathogens. Nevertheless, this is not a mandatory requirement. Final pathogen diagnosis was determined by two experienced clinicians based on epidemiology, clinical presentation, treatment outcome, laboratory findings, chest radiology findings, and the host's immune status, using a combination of mNGS and CMTs clinical diagnostic criteria. We will also review the original mNGS detection data if necessary.

We have summarized a table detailing clinical diagnosed pathogens per case and their relative abundance detected by mNGS. Please see Table S1 in the supplementary materials.

- Line 147 - underlying diseases in 19 patients is only mentioned in the results, the need or correlation of pathogens with comorbidities is unclear. I suggest to remove that or move the table to supplementary data.

Authors' response:

Thank you for your suggestion. We have removed the table. The information of 19 patients with underlying diseases were integrated together in TableS1 in the supplementary materials.

- Antibiotic adjustment is quite striking and would merit further discussion and highlight with a table to make it easier to follow.

Authors' response:

Thank you for the comment. The findings of 30 cases with antibiotic adjustment after pathogen diagnosis were summarized in Table S1. Details regarding antibiotic adjustments referred to pathogen diagnosis have been moved to section 3.4 of the Results (Page 6, Lines 208-219).

- Line 185 - I find confusing the statement that "specificity decreased by" for mNGS.

Please clarify - what was the specificity calculated from? Clinical diagnosis? from the originally defined 30% threshold? Conventional methods?

Authors' response:

Thank you for the comment and we are sorry for the confusion. The detection performance results were rewritten (Lines 192-207). The clinical pathogen diagnosis was used as the reference standard to calculate the sensitivity, specificity, positive predictive value (PPV), negative predictive value (NPV), and accuracy of mNGS and CMTs, respectively. The mNGS was performed using the reported results, while the CMTs were a combination of smear, acid-ast stain (AFS), BALF culture, multiplex PCR (mPCR), blood MP IgM and IgG antibody titres, blood tuberculosis infection T-cell spot (T-spot), and galactomannan (GM) assay results.

- The findings of resistance genes thanks to mNGS should be also highlighted, as it was for MP on line 217.

Authors' response:

Thank you for the comment. For the clinic, MP mutation information is a relatively clear marker to drug resistance. The findings of MP resistance gene mutation detected by mNGS have been added in the results (Page 5, Lines 187-191).

- Material and methods should include specifically what is included in each CMT described here. Line 245 states that one case failed to detect Aspergillus while galactomannan and "Aspergillus antigen" were detected. What antigen test is that?

Was that case confirmed to be Aspergillosis and treated like for it?

Authors' response:

Thank you for the comment. We apologize for the confusion caused by the description of the *Aspergillus* antigen. It has been confirmed that the patient (case 104 in Table S1) tested positive for the G test at an external hospital and positive for the GM test at our hospital. Additionally, the patient has responded effectively to Voriconazole treatment, leading to a diagnosis of *Aspergillus* infection. (Page 7, Lines 259-262)

Considering that the G test was conducted at outer hospital and was not routinely performed in this study at our hospital, it was not included in the CMTs in this study and only mentioned in the discussion as one of the supportive basis for the clinical diagnosis of *Aspergillus* infection in this case.

“In our hospital, a patient (case 104 in Table S1) who tested negative for mNGS and culture, was diagnosed with *Aspergillus* infection by the GM test and 1,3-β-D glucan (G) test (outer hospital testing), and showed a notable improvement in response to voriconazole antifungal therapy.”

- Line 248 - what is "G test"

Authors' response:

Thank you for the comment. G test is 1,3-β-D glucan test. We have revised this description (Page 7, Lines 261).

- Line 265 - mNGS shouldn't be compared to detecting IgM titers without adding the context of the biology of IgM, IgM should be discussed together with IgG as IgM is short lived and/or take immunity take longer to mount in an acute infection. Was that MP mNGS positive case, IgM negative also PCR negative?

Authors' response:

Thank you for the comment. We agree with your considerations for the interpretation of IgM in the context of its biological characteristics. Discussion on mNGS and IgM have been added in the discussion section (Page 8, Lines 299-302).

In this study, mNGS, IgM, and antibody titer were all tested in the acute phase of the infection. These different methods were used to independently and synchronously detect MP. This may be why mNGS had a higher positivity rate than IgM. When interpreting IgM results, we need to consider the time window of antibody production. We need other methods such as nucleic acid tests to provide evidence as well.

There were 17 cases that of MP mNGS positive and IgM negative. Among them, nine cases were mPCR (MP) positive, one case was mPCR (MP) negative, and mPCR (MP) was not available in seven cases.

- Data availability statement states "Data not publicly available", why is it not? NGS sequences could have been made available.

Authors' response:

Thank you for the comment. We have added the NGS data information in "Data availability statement" section (Page 9 Lines 347-350, Page 4, Lines 146-149).

Minor issues:

- Line 62 - correct "form" to "from"

Authors' response:

Thank you for the feedback. We have corrected it to the "from" in the revised manuscript (Page 2, Line 66).

- Line 161 - remove cursive on Cytomegalovirus humanbeta 5

Authors' response:

Thank you for the feedback. We have removed cursive on Cytomegalovirus humanbeta 5 (Page 5, Line 168).

- Consider moving Table 2 to supplementary data

Authors' response:

Thank you for your suggestion. We have removed the table 2. The information of 19 patients with underlying diseases were integrated together in TableS1 in the supplementary materials.

Reviewer #2 (Comments for the Author):

The manuscript by Zhao et al provides interesting information on the diagnostic value of mNGS in the context of children with severe pneumonia by reporting the results of mNGS and CMT in BALF for infection diagnosis in a retrospective cohort of 127 children. The manuscript is clearly written.

I have a few suggestions to improve the manuscript:

Authors' response:

We sincerely appreciate the insightful feedback. Please see below the point-by-point responses to your Comments.

Main:

L100: please specify the databases that have been used since kraken2 and bracken can be used with different databases

Authors' response:

Thank you for the comment. The database used in the study was added. Please see below and the revised manuscript for details (**Page 3, Lines 103-106**).

“The microorganism genome database containing genomes or scaffolds of bacteria, fungi, viruses, and parasites were downloaded from GenBank (release 238, <ftp://ftp.ncbi.nlm.nih.gov/genomes/genbank/>).”

L117: can you provide examples when a 30% cut off is not required beside

Mycobacterium

Authors' response:

Thank you for the comment. The criteria of clinically significant microorganisms (CSM) were employed to assist clinicians in the initial screening of mNGS-reporting pathogens. Nevertheless, this is not a mandatory requirement. Final pathogen diagnosis was determined by two experienced clinicians based on epidemiology, clinical presentation, treatment outcome, laboratory findings, chest radiology findings, and the host's immune status, using a combination of mNGS and CMTs clinical diagnostic criteria. If necessary, we will also review the original mNGS detection data if necessary.

We have summarized a table detailing clinical diagnosed pathogens per case and their relative abundance detected by mNGS. Please see **Table S1** in the supplementary materials. Case 1, 14,18, 52,59-60,66-69,72,80, 87,91,93,96,98,100,102,111 was examples when a 30% cut off is not required beside *Mycobacterium*.

L117: In fact, Mycobacterium spp (non-tuberculous species) are possible contaminants in the environment

Authors' response :

Thank you for your comment. No template negative controls (NTCs) were included in the extraction, library preparation, and sequencing process. It was used to exclude the process background contamination when reporting the mNGS results. The contamination from sampling was difficult to classify. Final pathogen diagnosis was determined by two experienced clinicians based on epidemiology, clinical presentation, treatment outcome, laboratory findings, chest radiology findings, and the host's immune status, using a combination of mNGS and CMTs clinical diagnostic criteria. If necessary, we will also review the original mNGS detection data if necessary.

We have revised the criteria (2) and added the criteria (3) to exclude possible contamination. Please see below and the revised manuscript for details (Page 4, Line 122-129).

“(2) Owing to the low wall-breaking extractions rate of Mycobacterium, if the number of strictly aligned sequences at the species level was ≥ 1 and contamination was ruled out using NTCs, the microorganism was classified as a CSM.”

“(3) Microorganisms detected in the NTCs were excluded unless the sequence number was ≥ 10 -fold higher than that in the NTCs.”

L201: the pathogenicity of *H. parainfluenzae* is very low while it is a very common colonizer belonging to the respiratory microbiota DOI: 10.1128/msystems.00178-21.

Can you discuss the likelihood of infection with *H. parainfluenzae* in your patients ?

Authors' response:

Thank you for your comment. The pathogenicity of *H. parainfluenzae* has been discussed in the Discussion section (Page 7, Line 282-289). *H. parainfluenzae* is an opportunistic pathogen that has been associated with endocarditis, COPD, otitis media, and, in rare cases, brain abscesses (PMID: 28674033, PMID: 31388489). Although *H. parainfluenzae* was often considered a member of the normal flora, it may be the pathogen responsible for the disease in immunocompromised patients without other evidence of etiology, especially when antibiotics are not effective (PMID: 36046746). In this study *H. parainfluenzae* was also found to be pathogenic according the improvement after antibiotic adjustment and in another study (PMID: 36169415). In some cases, the pathogenicity presumption was based on the detection of active expression in both DNA and RNA. This suggests that we also need to analyze mNGS results for rare pathogens in conjunction with CMTs and the patient's immune status to avoid over-interpretation.

L210-211: can you provide a timeline (or equivalent) of the detected MP to see if there is a link with the season which is an interesting information to guide the use of diagnosis tools

Authors' response:

Thank you for your comment. This study was conducted between June 2021 and March 2022, spanning nearly nine months, and included only children with severe or refractory pneumonia. Use this data to analyze the timeline of MP detection, and look for any seasonal correlations, as there may be potential biases.

L231: please be more cautious (at least except Mycobacteria)

Authors' response:

Thank you for your comment. A single case of *Mycobacterium tuberculosis* infection was identified in this study cohort. The *M. tuberculosis* case was detected by both mNGS and CMTs, and therefore *Mycobacterium tuberculosis* was not excluded in the calculation of bacteria (MP-) performance parameters such as specificity.

In the case of most bacteria, a negative result can be considered to exclude infection. However, in the case of *Mycobacterium tuberculosis*, the difficulty of wall-breaking can easily lead to false-negative results of the mNGS test. These results should be interpreted in conjunction with other factors such as clinical manifestations and laboratory tests.

Please see below and we have revised in the manuscript (Page 7, Line 245-250).

“It means that a negative mNGS result can be considered to exclude infection in the case of most bacteria. While in the case of *Mycobacterium tuberculosis*, the difficulty of wall-breaking can easily lead to false-negative results of the mNGS test. These results should be interpreted in conjunction with other factors such as clinical manifestations and laboratory tests.”

L238: "tricky" organisms: cultivating *S. pneumoniae* and *H. influenzae* is definitively less tricky than performing mNGS and allow antibiotic susceptibility testing. I think this really cannot be a justification for the mNGS use...

Authors' response:

Thank you for your comment. We are sorry for the confusion. "tricky" here mean the culture positive rates for *S. pneumoniae* and *H. influenzae* are relatively lower than mNGS, rather than the complexity of method operation. We have reorganized this discussion (Page 7, Line 253-256).

“Traditional bacterial cultures often yield low positive rates and are time- consuming. Some studies have demonstrated that mNGS can enhance the detection rate of a wide range of bacteria, including *Streptococcus pneumoniae*, *Haemophilus influenzae*, and *Escherichia coli* (Wei et al. 2023, Wu et al. 2020). This is beneficial for early diagnosis and treatment.”

L290: tNGS: if possible add a reference

Authors' response:

Thank you for the comment. tNGS references have been added (Page 8, Line326).

Minor:

L21: "retrospective" instead of "retrospectively"

Authors' response:

Thank you for the feedback. We have changed "retrospectively" to "retrospective"(Page 1, Line 21).

L25: add "by combining" mNGS and CMTs

Authors' response:

Thank you for your suggestion. We have added the “combining” between “by” and “mNGS and CMTs” (Page 1, Line25).

L31: add "compared to"

Authors' response:

Thanks for the comment. We have rewritten this sentence, as you suggested (Page2, Line 31).

L41: change by despite active management including anti-infection treatments

Authors' response:

Thanks for the comment. We have revised this sentence, as you suggested (Page 2, Line 48-49).

L47: you could add the following reference on mNGS use

10.1016/S2666-5247(23)00244-6

Authors' response:

Thanks for the comment. We have added this reference on mNGS, as you suggested (Page 2, Line 54)

L62: "from" instead of "form"

Authors' response:

Thank you for the feedback. We have corrected it to the “from” in the revised manuscript (Page 2, Line 66).

L121: should be "infecting, colonizing or contaminating"

Authors' response:

Thank you for your comment. We have revised this sentence, as you suggested (Page 4, Line 128).

L151: delete "rate of 100%"

Authors' response:

Thank you for your comment. We have deleted "rate of 100%", as you suggested (Page 6, Line 216-217).

L151: "for 1 patient" instead of "rate of 50%"

Authors' response:

Thank you for your comment. We have changed "rate of 50%" to “for one patient” in the revised manuscript (Page 6, Line 219).

Fig 3: please use italic when appropriate

Authors' response:

Thank you for your comment. We have updated the Figure 3 using italic font.

Re: Spectrum01087-24R1 (Diagnostic value of metagenomic next-generation sequencing using bronchoalveolar lavage fluid samples for pathogen detection in children with severe or refractory pneumonia)

Dear Dr. Meixi Zhao:

Thank you for the privilege of reviewing your work. Below you will find my comments, instructions from the Spectrum editorial office, and the reviewer comments.

You have addressed all the questions raised by the reviewers. However, there are still a few modifications needed before final acceptance for publication.

Page 7, Line 245-250, please consider the following text instead that proposed in the rebuttal letter

A negative mNGS result can generally be considered reliable for excluding infections from most bacteria. However, for *Mycobacterium tuberculosis*, the challenge of breaking cell wall can result in false-negative mNGS test outcomes. Therefore, these results should be interpreted alongside other factors, such as clinical manifestations and laboratory tests.

Line 179, please write as follows Human betaherpesvirus 5 (cytomegalovirus) to refer to Cytomegalovirus, and then use Cytomegalovirus.

The following text in the discussion remains confusing: the first sentence mentions a lower sensitivity of mNGS for *Mycoplasma pneumoniae* (MP), while the second sentence focuses solely on *Mycobacterium tuberculosis*.

For bacteria except MP, mNGS showed higher sensitivity (100.00%) and accuracy (77.95%) than CMTs, with a PPV and NPV of 68.18% and 100.00%, respectively. It means that a negative mNGS result can be considered to exclude infection in the case of most bacteria except *Mycobacterium tuberculosis* NGS has a good screening effect on bacterial infections. While in the case of *Mycobacterium tuberculosis*, the difficulty of wall-breaking can easily lead to false-negative results of the mNGS test. These results should be interpreted in conjunction with other factors such as clinical manifestations and laboratory tests. It is also crucial that the test specimen is representative of the site of infection.

Line 281. Reconsider the sentence "Traditional bacterial cultures often yield low positive rates and are time-consuming." as the time required for culture is usually less than that for mNGS.

The authors must provide a footnote for Table S1.

Revision Guidelines

Sincerely,
Jérôme Le Goff
Editor
Microbiology Spectrum

Dear Editor Jérôme Le Goff,

Thank you for reviewing our manuscript and allowing us to submit a revised manuscript entitled “Diagnostic value of metagenomic next-generation sequencing using bronchoalveolar lavage fluid samples for pathogen detection in children with severe or refractory pneumonia” (Spectrum01087-24) for further consideration by Microbiology Spectrum.

The authors sincerely appreciate the dedicated time and effort from the reviewers and editors on our manuscript. We are grateful for the insightful comments and have incorporated the reviewers’ suggestions into the revised manuscript. Please refer to the “Response to Reviewers”, in blue, to comments and concerns. We have uploaded the revised main manuscript - “Marked-Up Manuscript” file, where the changes are marked in red and as tracked changes. We also uploaded a clean version of the revised manuscript which convenient for you to browse and proofread the line numbers in this reply.

Thank you for your time and consideration! Please feel free to let us know if you need any further information.

Sincerely,

Meixi Zhao

Jinshan Hospital, Fudan University, Shanghai, China.

Email: 1342940100@qq.com

Point-by-Point Responses

DATE:2024-7-03

Response to Reviewer 1

Reviewer #1 (Public repository details (Required)):

Next gen sequencing information could have been deposited publicly.

Authors' response:

Thank you for the comment. We have added the NGS data information in "Data availability statement" section (Page 9 Lines 347-350) and in a "Data availability" paragraph placed at the end of the Materials and Methods section (Page 4, Lines 146-149). Please see below and the revised manuscript for details.

"Raw data of 127 cases were deposited in the National Genomics Data Center under project PRJCA028177 (GSA: CRA017789) and are publicly accessible at <https://ngdc.cncb.ac.cn/search/all?&q=PRJCA028177>."

Reviewer #1 (Comments for the Author):

Major issues:

- How is the threshold of 30% used validated? It would be good to include a table detailing those findings per case - can be in the supplementary section.

Authors' response:

Thank you for your suggestion. The threshold of 30% used here was referred to previous study (PMID: 35431561), that was already cited in the text. Interpreting mNGS reports has always been a challenge for clinicians. Here the criteria of

clinically significant microorganisms (CSM) were employed to assist clinicians in the initial screening of mNGS-reporting pathogens. Nevertheless, this is not a mandatory requirement. Final pathogen diagnosis was determined by two experienced clinicians based on epidemiology, clinical presentation, treatment outcome, laboratory findings, chest radiology findings, and the host's immune status, using a combination of mNGS and CMTs clinical diagnostic criteria. We will also review the original mNGS detection data if necessary.

We have summarized a table detailing clinical diagnosed pathogens per case and their relative abundance detected by mNGS. Please see Table S1 in the supplementary materials.

- Line 147 - underlying diseases in 19 patients is only mentioned in the results, the need or correlation of pathogens with comorbidities is unclear. I suggest to remove that or move the table to supplementary data.

Authors' response:

Thank you for your suggestion. We have removed the table. The information of 19 patients with underlying diseases were integrated together in TableS1 in the supplementary materials.

- Antibiotic adjustment is quite striking and would merit further discussion and highlight with a table to make it easier to follow.

Authors' response:

Thank you for the comment. The findings of 30 cases with antibiotic adjustment after pathogen diagnosis were summarized in Table S1. Details regarding antibiotic adjustments referred to pathogen diagnosis have been moved to section 3.4 of the

Results (Page 6, Lines 208-219).

- Line 185 - I find confusing the statement that "specificity decreased by" for mNGS.

Please clarify - what was the specificity calculated from? Clinical diagnosis? from the originally defined 30% threshold? Conventional methods?

Authors' response:

Thank you for the comment and we are sorry for the confusion. The detection performance results were rewritten (Lines 192-207). The clinical pathogen diagnosis was used as the reference standard to calculate the sensitivity, specificity, positive predictive value (PPV), negative predictive value (NPV), and accuracy of mNGS and CMTs, respectively. The mNGS was performed using the reported results, while the CMTs were a combination of smear, acid-ast stain (AFS), BALF culture, multiplex PCR (mPCR), blood MP IgM and IgG antibody titres, blood tuberculosis infection T-cell spot (T-spot), and galactomannan (GM) assay results.

- The findings of resistance genes thanks to mNGS should be also highlighted, as it was for MP on line 217.

Authors' response:

Thank you for the comment. For the clinic, MP mutation information is a relatively clear marker to drug resistance. The findings of MP resistance gene mutation detected by mNGS have been added in the results (Page 5, Lines 187-191).

- Material and methods should include specifically what is included in each CMT described here. Line 245 states that one case failed to detect *Aspergillus* while

galactomannan and "Aspergillus antigen" were detected. What antigen test is that?

Was that case confirmed to be Aspergillosis and treated like for it?

Authors' response:

Thank you for the comment. We apologize for the confusion caused by the description of the *Aspergillus* antigen. It has been confirmed that the patient (case 104 in Table S1) tested positive for the G test at an external hospital and positive for the GM test at our hospital. Additionally, the patient has responded effectively to Voriconazole treatment, leading to a diagnosis of *Aspergillus* infection. (Page 7, Lines 259-262)

Considering that the G test was conducted at outer hospital and was not routinely performed in this study at our hospital, it was not included in the CMTs in this study and only mentioned in the discussion as one of the supportive basis for the clinical diagnosis of *Aspergillus* infection in this case.

“In our hospital, a patient (case 104 in Table S1) who tested negative for mNGS and culture, was diagnosed with *Aspergillus* infection by the GM test and 1,3-β-D glucan (G) test (outer hospital testing), and showed a notable improvement in response to voriconazole antifungal therapy.”

- Line 248 - what is "G test"

Authors' response:

Thank you for the comment. G test is 1,3-β-D glucan test. We have revised this description (Page 7, Lines 261).

- Line 265 - mNGS shouldn't be compared to detecting IgM titers without adding the context of the biology of IgM, IgM should be discussed together with IgG as IgM is

short lived and/or take immunity take longer to mount in an acute infection. Was that MP mNGS positive case, IgM negative also PCR negative?

Authors' response:

Thank you for the comment. We agree with your considerations for the interpretation of IgM in the context of its biological characteristics. Discussion on mNGS and IgM have been added in the discussion section (Page 8, Lines 299-302).

In this study, mNGS, IgM, and antibody titer were all tested in the acute phase of the infection. These different methods were used to independently and synchronously detect MP. This may be why mNGS had a higher positivity rate than IgM. When interpreting IgM results, we need to consider the time window of antibody production.

We need other methods such as nucleic acid tests to provide evidence as well.

There were 17 cases that of MP mNGS positive and IgM negative. Among them, nine cases were mPCR (MP) positive, one case was mPCR (MP) negative, and mPCR (MP) was not available in seven cases.

- Data availability statement states "Data not publicly available", why is it not? NGS sequences could have been made available.

Authors' response:

Thank you for the comment. We have added the NGS data information in "Data availability statement" section (Page 9 Lines 347-350, Page 4, Lines 146-149).

Minor issues:

- Line 62 - correct "form" to "from"

Authors' response:

Thank you for the feedback. We have corrected it to the “from” in the revised manuscript (Page 2, Line 66).

- Line 161 - remove cursive on Cytomegalovirus humanbeta 5

Authors' response:

Thank you for the feedback. We have removed cursive on Cytomegalovirus humanbeta 5 (Page 5, Line 168).

- Consider moving Table 2 to supplementary data

Authors' response:

Thank you for your suggestion. We have removed the table 2. The information of 19 patients with underlying diseases were integrated together in TableS1 in the supplementary materials.

Reviewer #2 (Comments for the Author):

The manuscript by Zhao et al provides interesting information on the diagnostic value of mNGS in the context of children with severe pneumonia by reporting the results of mNGS and CMT in BALF for infection diagnosis in a retrospective cohort of 127 children. The manuscript is clearly written.

I have a few suggestions to improve the manuscript:

Authors' response:

We sincerely appreciate the insightful feedback. Please see below the point-by-point responses to your Comments.

Main:

L100: please specify the databases that have been used since kraken2 and bracken can be used with different databases

Authors' response:

Thank you for the comment. The database used in the study was added. Please see below and the revised manuscript for details (Page 3, Lines 103-106).

“The microorganism genome database containing genomes or scaffolds of bacteria, fungi, viruses, and parasites were downloaded from GenBank (release 238, <ftp://ftp.ncbi.nlm.nih.gov/genomes/genbank/>).”

L117: can you provide examples when a 30% cut off is not required beside Mycobacterium

Authors' response:

Thank you for the comment. The criteria of clinically significant microorganisms (CSM) were employed to assist clinicians in the initial screening of mNGS-reporting pathogens. Nevertheless, this is not a mandatory requirement. Final pathogen diagnosis was determined by two experienced clinicians based on epidemiology, clinical presentation, treatment outcome, laboratory findings, chest radiology findings, and the host's immune status, using a combination of mNGS and CMTs clinical diagnostic criteria. If necessary, we will also review the original mNGS detection data if necessary.

We have summarized a table detailing clinical diagnosed pathogens per case and their relative abundance detected by mNGS. Please see Table S1 in the supplementary

materials. Case 1, 14,18, 52,59-60,66-69,72,80, 87,91,93,96,98,100,102,111 was examples when a 30% cut off is not required beside *Mycobacterium*.

L117: In fact, *Mycobacterium* spp (non-tuberculous species) are possible contaminants in the environment

Authors' response :

Thank you for your comment. No template negative controls (NTCs) were included in the extraction, library preparation, and sequencing process. It was used to exclude the process background contamination when reporting the mNGS results. The contamination from sampling was difficult to classify. Final pathogen diagnosis was determined by two experienced clinicians based on epidemiology, clinical presentation, treatment outcome, laboratory findings, chest radiology findings, and the host's immune status, using a combination of mNGS and CMTs clinical diagnostic criteria. If necessary, we will also review the original mNGS detection data if necessary.

We have revised the criteria (2) and added the criteria (3) to exclude possible contamination. Please see below and the revised manuscript for details (Page 4, Line 122-129).

“(2) Owing to the low wall-breaking extractions rate of *Mycobacterium*, if the number of strictly aligned sequences at the species level was ≥ 1 and contamination was ruled out using NTCs, the microorganism was classified as a CSM.”

“(3) Microorganisms detected in the NTCs were excluded unless the sequence number was ≥ 10 -fold higher than that in the NTCs.”

L201: the pathogenicity of *H. parainfluenzae* is very low while it is a very common colonizer belonging to the respiratory microbiota DOI: 10.1128/msystems.00178-21.

Can you discuss the likelihood of infection with *H. parainfluenzae* in your patients ?

Authors' response:

Thank you for your comment. The pathogenicity of *H. parainfluenzae* has been discussed in the Discussion section (Page 7, Line 282-289). *H. parainfluenzae* is an opportunistic pathogen that has been associated with endocarditis, COPD, otitis media, and, in rare cases, brain abscesses (PMID: 28674033, PMID: 31388489). Although *H. parainfluenzae* was often considered a member of the normal flora, it may be the pathogen responsible for the disease in immunocompromised patients without other evidence of etiology, especially when antibiotics are not effective (PMID: 36046746). In this study *H. parainfluenzae* was also found to be pathogenic according the improvement after antibiotic adjustment and in another study (PMID: 36169415). In some cases, the pathogenicity presumption was based on the detection of active expression in both DNA and RNA. This suggests that we also need to analyze mNGS results for rare pathogens in conjunction with CMTs and the patient's immune status to avoid over-interpretation.

L210-211: can you provide a timeline (or equivalent) of the detected MP to see if there is a link with the season which is an interesting information to guide the use of diagnosis tools

Authors' response:

Thank you for your comment. This study was conducted between June 2021 and March 2022, spanning nearly nine months, and included only children with severe or refractory pneumonia. Use this data to analyze the timeline of MP detection, and look for any seasonal correlations, as there may be potential biases.

L231: please be more cautious (at least except Mycobacteria)

Authors' response:

Thank you for your comment. A single case of *Mycobacterium tuberculosis* infection was identified in this study cohort. The *M. tuberculosis* case was detected by both mNGS and CMTs, and therefore *Mycobacterium tuberculosis* was not excluded in the calculation of bacteria (MP-) performance parameters such as specificity.

In the case of most bacteria, a negative result can be considered to exclude infection. However, in the case of *Mycobacterium tuberculosis*, the difficulty of wall-breaking can easily lead to false-negative results of the mNGS test. These results should be interpreted in conjunction with other factors such as clinical manifestations and laboratory tests.

Please see below and we have revised in the manuscript (Page 7, Line 245-250).

“It means that a negative mNGS result can be considered to exclude infection in the case of most bacteria. While in the case of *Mycobacterium tuberculosis*, the difficulty of wall-breaking can easily lead to false-negative results of the mNGS test. These results should be interpreted in conjunction with other factors such as clinical manifestations and laboratory tests.”

L238: "tricky" organisms: cultivating *S. pneumoniae* and *H. influenzae* is definitively less tricky than performing mNGS and allow antibiotic susceptibility testing. I think this really cannot be a justification for the mNGS use...

Authors' response:

Thank you for your comment. We are sorry for the confusion. "tricky" here mean the culture positive rates for *S. pneumoniae* and *H. influenzae* are relatively lower than mNGS, rather than the complexity of method operation. We have reorganized this discussion (Page 7, Line 253-256).

“Traditional bacterial cultures often yield low positive rates and are time- consuming. Some studies have demonstrated that mNGS can enhance the detection rate of a wide range of bacteria, including *Streptococcus pneumoniae*, *Haemophilus influenzae*, and *Escherichia coli* (Wei et al. 2023, Wu et al. 2020). This is beneficial for early diagnosis and treatment.”

L290: tNGS: if possible add a reference

Authors' response:

Thank you for the comment. tNGS references have been added (Page 8, Line326).

Minor:

L21: "retrospective" instead of "retrospectively"

Authors' response:

Thank you for the feedback. We have changed "retrospectively" to "retrospective"(Page 1, Line 21).

L25: add "by combining" mNGS and CMTs

Authors' response:

Thank you for your suggestion. We have added the “combining” between “by” and “mNGS and CMTs” (Page 1, Line25).

L31: add "compared to"

Authors' response:

Thanks for the comment. We have rewritten this sentence, as you suggested (Page2, Line 31).

L41: change by despite active management including anti-infection treatments

Authors' response:

Thanks for the comment. We have revised this sentence, as you suggested (Page 2, Line 48-49).

L47: you could add the following reference on mNGS use

10.1016/S2666-5247(23)00244-6

Authors' response:

Thanks for the comment. We have added this reference on mNGS, as you suggested (Page 2, Line 54)

L62: "from" instead of "form"

Authors' response:

Thank you for the feedback. We have corrected it to the “from” in the revised manuscript (Page 2, Line 66).

L121: should be "infecting, colonizing or contaminating"

Authors' response:

Thank you for your comment. We have revised this sentence, as you suggested (Page 4, Line 128).

L151: delete "rate of 100%"

Authors' response:

Thank you for your comment. We have deleted "rate of 100%", as you suggested (Page 6, Line 216-217).

L151: "for 1 patient" instead of "rate of 50%"

Authors' response:

Thank you for your comment. We have changed "rate of 50%" to “for one patient” in the revised manuscript (Page 6, Line 219).

Fig 3: please use italic when appropriate

Authors' response:

Thank you for your comment. We have updated the Figure 3 using italic font.

DATE:2024-10-26

Dear Dr. Meixi Zhao:

Thank you for the privilege of reviewing your work. Below you will find my comments, instructions from the Spectrum editorial office, and the reviewer comments.

You have addressed all the questions raised by the reviewers. However, there are still a few modifications needed before final acceptance for publication.

1. Page 7, Line 245-250, please consider the following text instead that proposed in the rebuttal letter

A negative mNGS result can generally be considered reliable for excluding infections from most bacteria. However, for *Mycobacterium tuberculosis*, the challenge of breaking cell wall can result in false-negative mNGS test outcomes. Therefore, these results should be interpreted alongside other factors, such as clinical manifestations and laboratory tests.

Authors' response:

Thank you for your comment. We have replaced the description in that paragraph.

(Page 7, Line 245-250)

2. Line 179, please write as follows Human betaherpesvirus 5 (cytomegalovirus) to refer to Cytomegalovirus, and then use Cytomegalovirus.

Authors' response:

Thank you for your comment. We have modified the wording. **(Line 168)**

3.The following text in the discussion remains confusing: the first sentence mentions a lower sensitivity of mNGS for *Mycoplasma pneumoniae* (MP), while the second sentence focuses solely on *Mycobacterium tuberculosis*.

For bacteria except MP, mNGS showed higher sensitivity (100.00%) and accuracy (77.95%) than CMTs, with a PPV and NPV of 68.18% and 100.00%, respectively. It means that a negative mNGS result can be considered to exclude infection in the case of most bacteria except *Mycobacterium tuberculosis* NGS has a good screening effect on bacterial infections. While in the case of *Mycobacterium tuberculosis*, the difficulty of wall-breaking can easily lead to false-negative results of the mNGS test. These results should be interpreted in conjunction with other factors such as clinical manifestations and laboratory tests. It is also crucial that the test specimen is representative of the site of infection.

Authors' response:

Please see below and we have revised in the manuscript (Page 7, Line 245-250)

For bacteria except MP, mNGS showed higher sensitivity (100.00%) and accuracy (77.95%) than CMTs, with a PPV and NPV of 68.18% and 100.00%, respectively. A negative mNGS result can generally be considered reliable for excluding infections from most bacteria. However, for *Mycobacterium tuberculosis*, the challenge of breaking cell wall can result in false-negative mNGS test outcomes. Therefore, these results should be interpreted alongside other factors, such as clinical manifestations and laboratory tests. It is also crucial that the test specimen is representative of the site of infection.

4.Line 281. Reconsider the sentence "Traditional bacterial cultures often yield low positive rates and are time- consuming." as the time required for culture is usually less than that for mNGS.

Authors' response:

Thank you for your comment. We have modified the description in that paragraph.

(Line 254)

5.The authors must provide a footnote for Table S1.

Authors' response:

Thank you for your comment. We have added a footnote for Table S1.

Re: Spectrum01087-24R2 (Diagnostic value of metagenomic next-generation sequencing using bronchoalveolar lavage fluid samples for pathogen detection in children with severe or refractory pneumonia)

Dear Dr. Meixi Zhao:

Your manuscript has been accepted, and I am forwarding it to the ASM production staff for publication. Your paper will first be checked to make sure all elements meet the technical requirements. ASM staff will contact you if anything needs to be revised before copyediting and production can begin. Otherwise, you will be notified when your proofs are ready to be viewed.

Sincerely,
Jérôme Le Goff
Editor
Microbiology Spectrum